

# Sleep-wake patterns of fencing athletes: a long-term wearable device study

Jiansong Dai[1,*], Xiaofeng Xu[2,*], Gangrui Chen[3], Jiale Lv[2] and Yang Xiao[2]

[1] School of Sports and Health, Nanjing Sport Institute, Nanjing, Jiangsu, China
[2] Department of Graduate, Nanjing Sport Institute, Nanjing, Jiangsu, China
[3] Sport Science Research Institute, Nanjing Sport Institute, Nanjing, Jiangsu, China
[*] These authors contributed equally to this work.

## ABSTRACT

**Objective**. Sleep is the most efficient means of recovery for athletes, guaranteeing optimal athletic performance. However, many athletes frequently experience sleep problems. Our study aims to describe the sleep-wake patterns of fencing athletes and determine whether factors, such as sex, competitive level and training schedules, could affect the sleep-wake rhythm.

**Methods**. Sleep data from 23 fencing athletes were collected using the Huawei Band 6, monitoring key sleep parameters such as bedtime, wake time, duration of deep and light sleep, wake periods, REM sleep duration, and nap duration. During this period, athletes were required to wear the band continuously for 24 hours daily, except bathing, charging, and competition times.

**Results**. Athletes averaged 7.97 hours of sleep per night, with significant differences observed in wake time ($p = 0.015$) and midpoint of sleep ($p = 0.048$) between high-level and low-level athletes, as well as a higher frequency of naps among high-level ($\chi 2 = 11.97$, $p = 0.001$) and female ($\chi 2 = 3.88$, $p = 0.049$) athletes. Nap duration was negatively correlated with night sleep duration ($r = -0.270$, $p < 0.001$). Athletes were observed for changes in sleep-wake patterns from Monday to Sunday. On Mondays, Wednesdays, and Fridays, when there was no morning training, the athletes' wake-up time and the midpoint of sleep were shifted significantly backward, and there were significant differences in sleep parameters between training days and rest days.

**Conclusion**. The sleep patterns of athletes differ according to level and gender. The sleep-wake patterns of athletes are influenced by training schedules, indicating the presence of sleep rhythm disruption.

Corresponding author
Jiansong Dai, daijiansong@163.com

## INTRODUCTION

Sleep is a fundamental requirement for human health, playing a crucial role in promoting physical and psychological recovery (*Krueger et al., 2016*). For athletes, good sleep can enhance performance, facilitate fatigue recovery, and reduce the risk of injury (*Charest & Grandner, 2020*). In recent years, athletes' sleep has achieved increasing attention, but there is growing evidence to suggest that athletes do not obtain enough and high-quality sleep (*Gupta, Morgan & Gilchrist, 2017*; *Soligard et al., 2016*). Studies have shown that athletes

tend to sleep less on average than non-athletes (*Leeder et al., 2012*), and athletes' quality of sleep seems lower than their non-athlete peers (*Litwic-Kaminska & Jankowski, 2022*; *Mutsuzaki et al., 2018*).

A variety of factors influence athletes' sleep. The influence of sex hormones on circadian rhythms may contribute to differences in sleep between men and women (*Mong et al., 2011*). Women generally report more sleep problems than men, including athletes (*Goel, Kim & Lao, 2005*; *Schaal et al., 2011*; *Tsai & Li, 2004*). Sleep structure also varies among athletes of different levels, with elite athletes possibly having higher sleep efficiency (*Vlahoyiannis et al., 2021*). Due to the differences in the sports they engage in, athletes may also have varying sleep patterns. Previous studies have reported that athletes in individual sports tend to sleep less than those in team sports (*Leeder et al., 2012*; *Sargent et al., 2014*). *Aloulou et al. (2021)* found that subjective complaints related to sleep and somnolence were particularly high among fencing players, and soccer players. Sleep patterns may vary by gender, sport level, and sport type (*Hrozanova et al., 2021*; *Alves Facundo et al., 2022*). Current research indicates that more than half of athletes experience sleep problems on various occasions (*Gupta, Morgan & Gilchrist, 2017*; *Soligard et al., 2016*). The use of electronic devices, increased training volume and intensity, and the scheduling of training and competition may also reduce night sleep duration, impair sleep quality, and disrupt sleep rhythms (*Leatherwood & Dragoo, 2013*; *Surała et al., 2023*). For example, training scheduling may see an athlete sleep-in on the morning of a day off, before waking early on a training day (*Sargent et al., 2014*). This can lead to shortened sleep durations and potentially disrupt their sleep-wake rhythms. Furthermore, the impact of napping habits should also be considered, as they can significantly influence sleep patterns and recovery among athletes (*Lastella et al., 2021*). Napping can be a beneficial strategy for athletes to improve alertness, cognitive performance, and overall sleep quality, but it must be managed carefully to avoid interference with nighttime sleep (*Boukhris et al., 2023*; *Davies, Graham & Chow, 2010*).

Sleep issues experienced by athletes, such as insufficient sleep and disrupted sleep rhythms, can impact their athletic performance and potentially have adverse effects on their long-term health (*Simpson, Gibbs & Matheson, 2017*). Specifically, reduced and/or disturbed sleep has been shown to negatively influence aerobic (*Oliver et al., 2009*) and anaerobic performance metrics (*Skein et al., 2013*; *Souissi et al., 2008*), increase injury and illness risk (*Charest & Grandner, 2020*; *Milewski et al., 2014*). Insufficient sleep has also been demonstrated to exert negative impacts on attention and reaction time. Merely one night of total sleep deprivation can adversely affect reaction time (*Basner & Dinges, 2011*; *Milewski et al., 2014*). During competitions or events, this may alter an athlete's ability to make instantaneous and accurate decisions.

As far as we know there have been some sleep studies on soccer and rugby or swimmers and runners, but almost no studies on fencers' sleep alone. Fencing distinguishes itself from other sports as an individual and hybrid discipline, requiring athletes to possess intricate offensive and defensive skills that encompass agile transitions, immediate stops, and instantaneous bursts of power. Furthermore, fencing necessitates a high degree of mental concentration and rapid reflexes from athletes. A single training session or competition

in fencing not only results in muscular fatigue but also induces psychological fatigue. Therefore, it is necessary to gain a thorough understanding of the sleep characteristics of fencing athletes in order to assist them in achieving recovery and maintaining optimal athletic performance.

Currently, there are many methods available for evaluating and monitoring athletes' sleep. Objective sleep assessment methods include the "gold standard" polysomnography (PSG) (*Jafari & Mohsenin, 2010*), actigraphy (*e.g.*, GT3X) (*Sargent et al., 2016*), and various commercial monitoring devices (*e.g.*, FitBit, smart bracelets) (*Vlahoyiannis et al., 2021*). Subjective sleep evaluation methods include sleep diaries and various sleep scale questionnaires, such as the Pittsburgh Sleep Quality Index (PSQI) (*Buysse et al., 1989*), the Athlete Sleep Screening Questionnaire (ASSQ) (*Samuels et al., 2016*), and the Athlete Sleep Behavior Questionnaire (ASBQ) (*Halson, 2019*). Different sleep evaluation methods can impact the results of sleep assessments. The most common method is the subjective sleep questionnaires, but it has been shown to conflict with objective sleep measurements. Although PSG is considered the gold standard for sleep measurement, the process usually requires a professional's presence in a sleep laboratory, which is inconvenient for athletes (*De Zambotti et al., 2019*; *Kwon, Kim & Yeo, 2021*). In previous studies, the monitoring of athletes' sleep duration typically did not exceed two weeks, with rarely conducted long-term sleep monitoring extending beyond one month. Such short-term sleep monitoring may be influenced by transient factors, failing to adequately capture individual sleep patterns and trends, and thus limiting comprehensive assessments of sleep quality. Therefore, long-term sleep monitoring of athletes is necessary. With advancements in science and technology, wearable smart devices that provide improved user experiences, ease of use, and rapid sleep feedback are increasingly utilized for the daily monitoring of athletes.

Our study aimed to investigate and analyze the sleep-wake patterns of athletes in the Jiangsu Provincial Fencing Team by having them wear smart bracelets to continuously monitor their sleep changes. The study focused on examining the sleep characteristics of athletes across different genders and sport levels, as well as the impact of training schedules on their sleep patterns. The aim is to assist fencing athletes in developing personalized training plans and lifestyles that optimize their athletic performance.

## MATERIALS & METHODS

### Participants

A total of 23 professional fencing athletes from the Jiangsu Province Fencing Team (Nanjing, China) were selected for monitoring. Details of the participants are provided in Table 1. Athletes who win competitions at or above the national level will be awarded "first-class athletes" or higher. In contrast, those who win competitions at or below the provincial level will be awarded "second-class athletes" or lower. Athlete grade certificates are issued by the General Administration of Sport of China based on competition performance, representing different competitive levels. This study was approved by the Ethics Committee of Nanjing Sport Institute (No.RT-2022-01). Prior to data collection, participants received a detailed information sheet about the purpose of the study and provided written informed consent.

**Table 1  Basic information of athletes.**

|  | Male | Female |
|---|---|---|
| n | 11 | 12 |
| Age (y) | 20.82 ± 2.68 | 21.42 ± 2.58 |
| Height (cm) | 184.82 ± 6.19 | 171.92 ± 5.52 |
| Weight (kg) | 75.45 ± 9.66 | 62.08 ± 6.22 |
| BMI (kg/cm$^2$) | 22.01 ± 1.69 | 21.01 ± 2.07 |
| Level | High (4) | High (7) |
|  | Low (7) | Low (5) |

## Measurement indicators

The Huawei band 6 was used to collect sleep and training data of 23 athletes from April to September 2023. During this period, athletes were required to wear the band continuously 24 h a day (except during bathing and charging time). The band was connected to a mobile phone *via* Bluetooth. Athletes uploaded the data to the HUAWEI Research cloud platform daily through a dedicated APP. The researchers then downloaded the data from the cloud platform and analyzed it. It is noteworthy that, given athletes' usual practice of not wearing such monitoring devices during official competitions, relevant data during competition periods were excluded from the data collection phase of this study to avoid data bias caused by differences in wearing habits. To ensure data continuity and completeness, researchers reminded participants daily to wear the bracelets. However, despite these measures, due to the extended monitoring period, there were inevitably some days where data were missing for various reasons (such as forgetfulness, device malfunction). Additionally, records with incomplete data were excluded from the uploads, resulting in a total of 2,459 days of valid sleep data collected.

The weekly training schedule was as follows: training sessions were held on Tuesday, Thursday, and Saturday mornings (9:00–11:00 a.m.) and afternoons (3:00–5:00 p.m.). On Monday, Wednesday, and Friday mornings, the athletes were engaged in cultural studies or resting, with almost no training scheduled, while training was conducted in the afternoons (3:00–5:00 p.m.). Sunday was a rest day. All the participants were from a sports team and received similar daily.

Sleep parameters obtained from the band included sleep onset time, wake-up time, deep sleep duration, light sleep duration, awake duration, REM duration, nap duration, and total sleep duration. Additionally, night sleep duration and mid-sleep point were calculated. Night sleep duration referred to the duration from sleep onset to wake-up time, excluding awake duration. The mid-sleep point was the midpoint between sleep onset and wake-up time. For example, if sleep onset occurred at 23:00 and wake-up time occurred at 07:00, the mid-sleep point would be at 04:00.

The Huawei wristband integrates a variety of sensors and algorithms to comprehensively monitor the user's physiological and movement status. The wristband primarily utilizes Photoplethysmography (PPG) (*Fonseca et al., 2017*) to monitor autonomic activities such as heart rate and pulse waves. Coupled with machine learning technology (*Fonseca et al., 2017*; *Reimer et al., 2017*), the wristband is capable of tracking sleep and wakefulness

states through motion tracking (*e.g.*, accelerometer) and autonomic activities monitoring (*Domingues, Paiva & Sanches, 2014*; *Fonseca et al., 2017*; *Reimer et al., 2017*). Notably, cardiopulmonary coupling (CPC) (*Thomas et al., 2005*) and heart rate variability analysis (*Domingues, Paiva & Sanches, 2014*), as cardiorespiratory sleep staging techniques, further enhance the accuracy of the wristband's sleep quality assessment. Collectively, these technologies enable the Huawei wristband to provide continuous, real-time health monitoring and data analysis for users. The accuracy of the wearable devices assessing sleep has been validated by comparing with in-lab video-polysomnography (*Thomas et al., 2005*; *Xie et al., 2018*).

## Statistical analysis

Data were processed and analyzed using Excel and SPSS 22.0. The descriptive results of measurement data were expressed as x $\pm$ s, and the frequency analysis was used for count data, expressed as percentages. A linear mixed model was used to compare differences in sleep-wake patterns across genders and sport levels, with "gender", "sport level" or "gender*sport level" as fixed terms and "athlete ID" as a random term. Differences in nap frequency among athlete types were compared using the chi-square test. Pearson's correlation was used to analyze the relationship between daytime naps and night sleep duration. To explore the effect of training schedule on athletes' sleep-wake patterns, sleep onset time, wake-up time, and mid-sleep point of athletes from Monday to Sunday were analyzed by repeated measure ANOVA. A linear mixed model was also used to compare the differences in sleep between fencers' rest days and training days, with "training day/rest day" as a fixed term and "athlete ID" as a random term. The significance level was set at $P < 0.05$ for all tests, and effect sizes were calculated using partial $\eta$ squared ($\eta$p2) and interpreted as small (0.01), medium (0.06), or large (0.14).

## RESULTS

### Characteristics of night sleep

On average, fencing athletes slept at 23:56 and woke up at 8:00, with an average night sleep duration of 478 min (7.97 h).

There were no significant differences in sleep parameters between male and female athletes, and Table 2 shows the sleep characterization of fencing athletes of different genders. The differences in wake-up time ($p = 0.015$, $F = 6.971$, $\eta p^2 = 0.330$) and midpoint of sleep ($p = 0.048$, $F = 4.405$, $\eta p^2 = 0.0209$) of athletes at different levels were statistically significant. In contrast, the others were not statistically significant, and the sleep characterization of the fencing athletes of the different levels of fencing is shown in Table 3. The analysis revealed that the interaction between gender and exercise level did not achieve statistical significance for parameters.

### Characteristics of napping

In our study, female athletes showed a significantly higher frequency of napping compared to male athletes ($\chi 2 = 3.88$, $p = 0.049$), and high-level athletes had a significantly higher napping frequency than low-level athletes ($\chi 2 = 11.97$, $p = 0.001$). There were

**Table 2 Sleep characteristics of fencers of different genders.**

|  | All ($n = 23$) | Male ($n = 11$) | Female ($n = 12$) |
|---|---|---|---|
| Sleep onset time (hh:mm) | 23:56 ± 1:21 | 00:09 ± 1:17 | 23:28 ± 1:21 |
| Wake-up time (hh:mm) | 8:00 ± 1:37 | 8:09 ± 0:55 | 8:06 ± 0:45 |
| Mid-point of sleep (hh:mm) | 3:53 ± 1:02 | 4:05 ± 1:18 | 3:43 ± 1:09 |
| Deep sleep duration (min) | 134.5 ± 40.6 | 128.7 ± 8.4 | 140.2 ± 27.3 |
| Light sleep duration (min) | 252.2 ± 66.7 | 243.9 ± 23.6 | 256.2 ± 39.1 |
| REM duration (min) | 92.0 ± 33.7 | 85.7 ± 9.6 | 94.7 ± 14.3 |
| Awake duration (min) | 16.2 ± 23.8 | 17.8 ± 6.9 | 14.7 ± 6.0 |
| Night sleep durtion (min) | 478.7 ± 96.3 | 456.0 ± 77.3 | 496.7 ± 105.6 |

**Table 3 Sleep characteristics of fencers of different levels.**

|  | High-level ($n = 11$) | Low-level ($n = 12$) |
|---|---|---|
| Sleep onset time (hh:mm) | 23:58 ± 1:31 | 23:35 ± 1:10 |
| Wake-up time (hh:mm) | 8:25 ± 1:48 | 7:39 ± 1:20[*] |
| Mid-point of sleep (hh:mm) | 4:11 ± 1:22 | 3:38 ± 1:02[*] |
| Deep sleep duration (min) | 141.0 ± 46.6 | 128.7 ± 33.5 |
| Light sleep duration (min) | 255.5 ± 68.5 | 249.3 ± 65.0 |
| REM duration (min) | 95.3 ± 34.8 | 89.1 ± 32.4 |
| Wwake duration (min) | 15.1 ± 21.1 | 17.2 ± 26.0 |
| Night sleep durtion (min) | 491.8 ± 111.1 | 467.2 ± 79.3 |

Notes.
[*] Significant difference between athletes of different levels ($p < 0.05$).

**Table 4 Napping behaviour for fencers.**

|  | Male | Female | High-level | Low-level |
|---|---|---|---|---|
| Total number of days of data collection (count) | 1088 | 1371 | 1150 | 1309 |
| Naps during data collection (count) | 670 | 897 | 774 | 793 |
| Days without naps during data collection (count) | 418 | 474 | 376 | 516 |
| Nap frequency (%) | 61.6 | 65.4[#] | 67.3 | 60.6[*] |
| Nap duration (min) | 59.3 ± 60.9 | 56.1 ± 54.7 | 63.4 ± 58.2 | 52.4 ± 56.5 |

Notes.
[#] Significant difference between athletes of different genders ($p < 0.05$).
[*] Significant difference between athletes of different levels ($p < 0.05$).

no significant differences in nap duration between genders or among fencing athletes of different levels. The napping patterns of fencing athletes are presented in Table 4. Pearson correlation analysis examined the relationship between daytime nap and night sleep duration, revealing a negative correlation between the two variables ($r = -0.256$, $p < 0.001$).

**Effects of training schedule on sleep-wake patterns in fencing athletes**

The results of the one-way repeated measures ANOVA showed significant differences in sleep onset time, wake-up time, and mid-point of sleep from Monday to Sunday ($F = 19.18, p < 0.001, \eta p2 = 0.217; F = 35.75, p < 0.001, \eta p2 = 0.383; F = 38.645, p < 0.001, \eta p2 = 0.405$). The specific changes are shown in Fig. 1. On training days, fencing athletes typically slept at 23:43 and woke up at 7:49, resulting in an average nightly sleep duration of 470 min (7.83 h). On rest days, they went to sleep at 00:22 and woke up at 9:42, resulting in an average nightly sleep duration of 548 min (9.13 h). Differences in sleep between training and rest days are detailed in Table 5.

# DISCUSSION

## Sleep sleep-wake patterns in fencing athletes: comparing sex and sport level

Research suggests that approximately 8 h of sleep per night can help prevent neurobehavioral deficits associated with sleep deprivation (*Belenky et al., 2003*). International guidelines also recommend that athletes require 8–10 h of sleep per night to facilitate physical and psychological recovery following intense training and competition (*Kellmann et al., 2018*). However, it is noteworthy that specific research on fencing athletes remains scarce. A survey examining the sleep habits of athletes across diverse sports revealed that fencing athletes, on average, slept for 414.2 min (6.9 h) during the night, markedly falling short of the sleep duration necessary for optimal recovery (*Aloulou et al., 2021*). In contrast, in our study, the average night sleep duration for fencing athletes was 7.97 h, slightly below the recommended 8-hour sleep standard. Several factors may contribute to these differing results. The participants in the study by Aloulou and colleagues were adolescents, and existing research has indicated that age has a significant impact on athletes' sleep patterns (*Fox et al., 2020*). Differences in athlete management models may also be another factor contributing to variations in sleep duration. Compared to athletes from other countries, Chinese athletes typically undergo a more unified and strictly regulated management system, which may exert different influences on their sleep habits. Furthermore, *Kamdar et al. (2004)* found that increasing habitual sleep duration from 7 to 8 h per night can significantly enhance reaction speed and reduce daytime sleepiness and fatigue levels, further emphasizing the importance of adequate sleep for athletic performance.

Therefore, fencing athletes who do not meet the 8-hour sleep standard, it is recommended to take proactive measures to increase their sleep duration. Fencing athletes' sleep habits revealed that their average bedtime was 23:56, indicating a relatively late bedtime. The biological clock habits of going to bed late, the stress associated with training and competition, and behaviors such as excessive use of electronic devices in the evening may cause athletes to fall asleep at a later time. Therefore sleep education for athletes should be enhanced to adjust the sleep onset time.

In our study, the average night sleep duration was 7.6 h for male athletes and 8.3 h for female athletes. The analysis indicated that these differences were not statistically significant, contradicting the findings of the previous study (*Roberts et al., 2022*). This discrepancy may

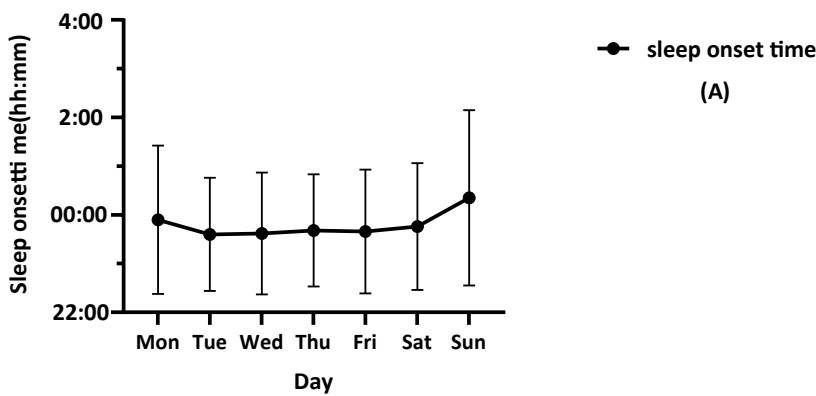

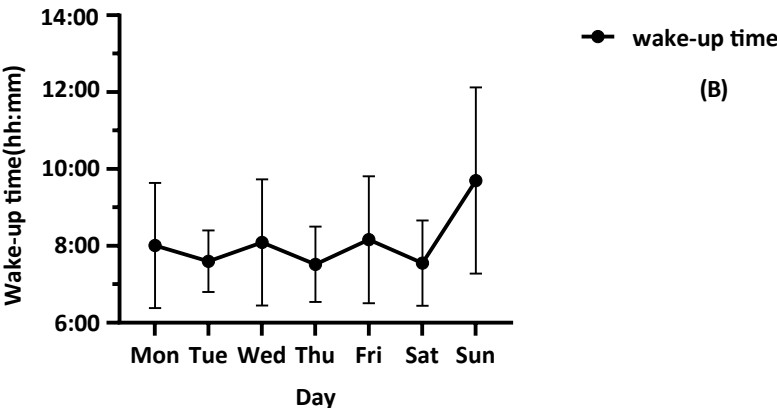

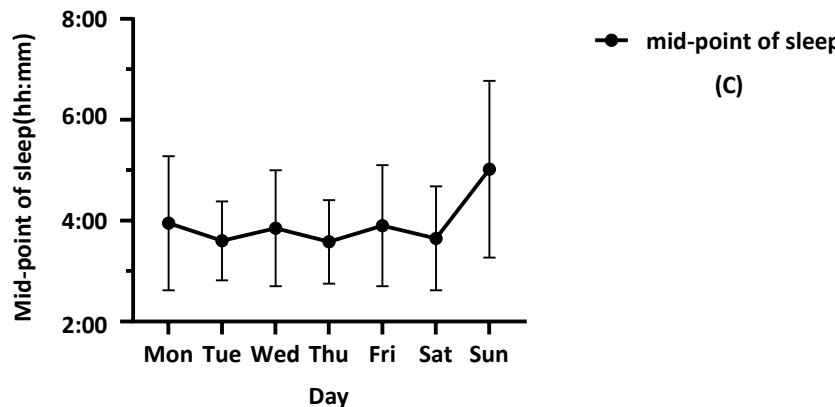

Figure 1 Changes in sleep-wake patterns in fencers from Monday to Sunday.

**Table 5  Differences in sleep between training days and rest days among fencers.**

|  | Training days ($n = 2256$) | Rest days ($n = 260$) |
|---|---|---|
| Sleep onset time (hh:mm) | 23:43 ± 1:17 | 0:22 ± 1:49[*] |
| Wake-up time (hh:mm) | 7:49 ± 1:22 | 9:42 ± 2:25[*] |
| Mid-point of sleep (hh:mm) | 3:45 ± 1:05 | 5:01 ± 1:45[*] |
| Deep sleep duration (min) | 133.2 ± 38.9 | 146.2 ± 51.4[*] |
| Light sleep duration (min) | 247.3 ± 59.8 | 294.2 ± 100.3[*] |
| REM duration (min) | 90.2 ± 32.3 | 107.6 ± 40.7[*] |
| Awake duration (min) | 16.5 ± 24.1 | 12.8 ± 21.1[*] |
| Night sleep durtion (min) | 470.7 ± 85.8 | 548.2 ± 142.7[*] |
| Nap duration (min) | 61.9 ± 57.4 | 19.4 ± 41.8[*] |
| Total sleep durtion (min) | 532.6 ± 90.9 | 567.6 ± 146.5[*] |

**Notes.**
Significant differences in sleep parameters between training days and rest days ($p < 0.05$).

stem from a combination of factors, including the limited sample size employed in our study and the unique training characteristics of elite athletes. Among elite athletes, the variability in sleep patterns tends to diminish due to their stringent training schedules and shared competitive environments. Despite this reduced variability, our study still identified an overall trend indicating better objective sleep quality among female athletes. Male and female athletes had comparable wake-up time due to similar training schedules, but female athletes generally fell asleep earlier. Male athletes were also awake longer on average than the female athletes and awakened more frequently during the night. Notably, despite the better objective sleep quality, subjective sleep quality was poorer among female athletes, with a 55% higher likelihood of experiencing sleep disorders compared to their male counterparts (*Benjamin et al., 2020*; *Hrozanova et al., 2021*). Although the exact causes remain unclear, it is suggested that the high prevalence of depression and anxiety (*Bruck & Astbury, 2012*), as well as ovarian steroid hormones (*Driver et al., 1996*), may be potential factors contributing to the increased risk of sleep problems in women.

There was a significant difference in morning wake-up time between high-level and low-level athletes. The athletes' age and the organization of the training schedule may be the primary factors that influence this. The regularity and competitive nature of fencing impose specific age-related requirements on athletes (*Zhang, 2005*). In our study, the average age of high-level athletes was 22.5 years, whereas the average age of low-level athletes was 19.9 years. There was a difference in age between the two groups of athletes. According to the Jiangsu Provincial Fencing Team's training schedule, younger athletes have school classes on Monday, Wednesday, and Friday mornings, necessitating earlier wake-up time for these athletes. In our study, low-level athletes also tended to have shorter night sleep duration and longer periods of wakefulness during the night. This may be because low-level athletes are in the late stages of adolescence or the early stages of adulthood, during which social and psychosocial factors can lead to more irregular sleep-wake patterns (*Carskadon, 2011*; *Leigh & Clark, 2018*). Our study shows that high-level fencing athletes tend to fall asleep later and wake up later, which is similar to the latest survey on Chinese athletes. This research suggests that even when age was used as a mediating variable, high-level Chinese

athletes still tended to be more evening oriented than lower-level athletes (*Tan et al., 2023*). Individual differences in athletes' chronotypes can impact their sleep patterns. High-level athletes may also face greater stress, which may lead to falling asleep later. Additionally, due to the physical expenditure incurred by intense training, they require a longer duration of sleep to facilitate recovery, further postponing their wake-up time and resulting in a longer sleep duration. Sleep is an essential factor in optimizing athletic performance and recovery, and the consequences of sleep deprivation are significant for elite athletes, especially young people (*Simpson, Gibbs & Matheson, 2017*). More research is needed to explore the relationship between sleep and sports performance. Specifically, whether good sleep habits enhance athletic performance or whether high-performing athletes naturally exhibit better sleep requires further investigation.

Napping is a widespread behaviour and a critical public health practice (*Faraut et al., 2017*). Athletes also use nap to increase their total sleep duration, making napping a regular part of many athletes' training programs (*Sargent et al., 2014*). The present study showed that fencing athletes who had shorter night sleep duration were more likely to take longer nap during the day. Furthermore, nap duration was significantly higher on training days than on rest days, reflecting athletes' tendency to supplement their sleep needs with daytime naps. Female athletes had a higher nap frequency than male athletes. Similarly, high-level athletes napped more frequently than those with low-level athletes, indicating that female athletes and high-level athletes tended to have a better habit of napping. The benefits of napping on athletic performance following sleep deprivation are well-documented. Naps improve mood, alertness, and reaction time and serve as an effective way to reduce daytime sleepiness, particularly for athletes who engage in morning training (*Lastella et al., 2021*). However, it is controversial whether longer nap duration is better. It has been suggested that when the nap duration is too long for more than one sleep cycle, it may affect the night sleep by influencing the sleep-wake pattern, resulting in shorter sleep duration and decreased efficiency at night (*Goldman et al., 2008*). In a study on Chinese athletes, *Liao et al. (2018)* showed that a 60 min nap was more favorable to shorten the night sleep latency and prolong the duration of deep sleep compared with a 120 min nap, and suggested that good athletes without sleep disorders and used to napping should choose a nap sleeping strategy that is shorter than the duration of a complete sleep cycle. For professional athletes, cultivating good sleep habits is particularly crucial. Athletes should have a profound understanding of the intrinsic relationship between nighttime sleep and daytime sleep needs, ensuring adequate sleep at night while reasonably managing nap duration, in order to continuously optimize and enhance their athletic performance.

## Irregular sleep-wake patterns in fencing athletes

Athletes' sleep is closely related to their training and competition schedules. *Sargent et al. (2012)* studied the sleep-wake patterns of seven Olympic swimmers and found that earlier training start times were associated with reduced sleep duration the preceding night. Furthermore, compared to the night preceding a rest day, athletes spent less time in bed and exhibited a shorter night sleep duration prior to training sessions (*Sargent et al., 2014*).

These findings collectively suggest that their daily training schedule directly influences athletes' sleep quality and quantity.

Our study observed significant variations in athletes' wake-up times and midpoints of sleep from Monday to Sunday, indicating a pronounced impact of their schedules on sleep rhythms. Specifically, on Mondays, Wednesdays, and Fridays, the athletes' sleep patterns changed because they did not have to wake up early to train as they did on other training days. Some athletes went to class while others rested, giving them a more flexible wake-up time. Therefore, it can be found that on Monday, Wednesday, and Friday, the athletes' midpoint of sleep and wake-up time are generally delayed.

Many people try to compensate for sleep on weekdays by sleeping longer on weekends. The sleep monitoring of fencing athletes also revealed this pattern, with athletes sleeping significantly longer on rest days than on training days. Compensatory sleep on rest days may help alleviate some sleep debt, but it does not fully mitigate the negative effects of sleep deprivation on physical and cognitive functions (*Depner et al., 2019*). Athletes' behaviors of staying up late and compensating for sleep on rest days can disrupt their sleep-wake patterns and lead to circadian rhythm disturbances, which in turn affect physiological rhythms (*Pradhan et al., 2024*). The variation in sleep patterns between training days and weekdays, known as social jet lag (SJL), is similar to traveling across multiple time zones and returning to the original time zone on Monday morning (*Gentry et al., 2021*). This change may leave athletes experiencing jet lag on Monday, impacting their training, performance, and emotional regulation (*Levandovski et al., 2011*). However, SJL differs from travel jetlag in that the latter is typically a one-time occurrence, while social jetlag is a recurring weekly phenomenon, making it a persistent and significant issue. *Cespedes Feliciano et al. (2019)* suggested that prolonged SJL can lead to issues such as diabetes, obesity, and cardiovascular diseases. Moreover, the more severe SJL, the greater the risk of adverse health, including cognitive dysfunction (*Taillard et al., 2021*), inattention, and mental disorders (*Okajima et al., 2021*).

The sleep onset times of athletes on training days showed no significant variation, but the morning schedules affected their wake-up times, which in turn influenced the night sleep duration. On rest days, both sleep onset times and wake-up times were significantly delayed. For fencers, athletes should adjust their routine to maintain a healthy and stable sleep rhythm. Fluctuations in wake-up times due to training schedules may disrupt their circadian rhythms. Therefore, athletes should go to bed earlier and maintain a regular wake-up time each day while ensuring adequate sleep. In recent years, the issue of sleep-wake (*Alves Facundo et al., 2022*; *Mong et al., 2011*) has increasingly become a focus of public attention. Researchers have emphasized that, while it is important to ensure athletes obtain sufficient sleep duration, maintaining their sleep-wake rhythm is equally crucial (*Okajima et al., 2021*; *Sletten et al., 2023*). Our research findings further reveal irregular sleep patterns among athletes. By conducting sleep monitoring, we can gain a deeper understanding of individual differences in athletes' sleep and accordingly propose targeted improvement strategies. In the future, it is recommended to adopt objective sleep assessment tools before conducting sleep education for athletes, in order to ensure the precision and effectiveness of the educational content.

## LIMITATIONS

Despite providing some meaningful results, our study has several limitations. Firstly, the sample size was relatively small, which may limit the generality and reliability of the findings. Future research with a larger sample size is needed to further validate our observations. Secondly, we collected data using Huawei wearable devices. While this method has certain accuracy and has been validated for use in adults, it lacks validation specifically among athletes. Therefore, the interpretation of the results should be approached with caution. Additionally, although all athletes followed similar training schedules, individual differences may lead to varied responses in sleep needs. For instance, some athletes may be more susceptible to the impact of training loads, resulting in poorer sleep quality. Future studies can explore the specific effects of different training programs on athletes' sleep.

## CONCLUSIONS

The sleep-wake patterns of athletes are significantly influenced by their training schedules, culminating in perturbations within their sleep-wake rhythm, a phenomenon that necessitates attention in the field of sports science and medicine. Also, sleep characteristics may differ by gender and sport level. Objective sleep monitoring provides valuable insights into these variations, aiding coaches in developing personalized training plans. Continuous monitoring and feedback also help athletes better understand their sleep patterns and implement necessary adjustments. Future research should increase sample sizes and utilize advanced sleep monitoring technologies to explore factors influencing sleep more comprehensively. Additionally, investigating the impact of sleep on athletic performance and overall well-being will enhance guidance for training and lifestyle optimization.

## ACKNOWLEDGEMENTS

This author would like to thank each of the athletes as well as the coaches for their invaluable participation and cooperation throughout the study.

### Funding

This study was supported by the ''2023 Major Sports Research Project of Jiangsu Province Sports Bureau'' (ST231205) by the Jiangsu Province Sports Bureau of China. The funders had no role in study design, data collection and analysis, decision to publish, or preparation of the manuscript.

### Grant Disclosures

The following grant information was disclosed by the authors:
The ''2023 Major Sports Research Project of Jiangsu Province Sports Bureau'': ST231205.

### Competing Interests

The authors declare there are no competing interests.

## Author Contributions

- Jiansong Dai conceived and designed the experiments, performed the experiments, analyzed the data, prepared figures and/or tables, authored or reviewed drafts of the article, and approved the final draft.
- Xiaofeng Xu conceived and designed the experiments, performed the experiments, analyzed the data, prepared figures and/or tables, and approved the final draft.
- Gangrui Chen performed the experiments, authored or reviewed drafts of the article, and approved the final draft.
- Jiale Lv performed the experiments, authored or reviewed drafts of the article, and approved the final draft.
- Yang Xiao performed the experiments, authored or reviewed drafts of the article, and approved the final draft.

## Human Ethics

The following information was supplied relating to ethical approvals (*i.e.*, approving body and any reference numbers):

This study was approved by the Ethics Committee of Nanjing Sport Institute (No.RT-2022-01).

## Data Availability

The raw measurements are available in the Supplemental File.

## Supplemental Information

Supplemental information for this article can be found online at http://dx.doi.org/10.7717/peerj.18812#supplemental-information.

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
