# Peer review of "Sleep-wake patterns of fencing athletes: a long-term wearable device study"

_PeerJ, doi:10.7717/peerj.18812_

## Round 0.1 · original submission · Major Revisions

Dear Authors:

I certainly have a lot of doubts about your manuscript titled: "Sleep-wake patterns of fencing athletes: a long-term wearable device study". Please respond to the reviewers (especially reviewer #2, who suggests rejection. I need to read your answers to their concerns). So, I decided on "major reviews" and will read your resubmitted manuscript.

Regards

Dr. Manuel Jiménez

Reviewer 1 ·

Basic reporting

The article is written in clear and well-structured English, with accurate use of technical terminology. The background provided in the introduction is appropriate and highlights the importance of the study on athletes' sleep. The experimental design is consistent and adequately described. The use of a wearable device (Huawei Band 6) for sleep monitoring represents a practical solution, given the challenges associated with using PSG in sports environments.
The results are clearly presented, and the use of appropriate statistical methods to compare group differences and correlations is well explained.
The conclusions align with the presented data and provide practical recommendations for future training programs.

Lines 19-25:
You have provided a good summary of the data collection methods. To ensure clarity, consider slightly condensing the details while maintaining the essential information.
Example: "Sleep data from 23 fencing athletes were collected using the Huawei Band 6, monitoring key sleep parameters such as bedtime, wake time, duration of deep and light sleep, wake periods, REM sleep duration, and nap frequency."

Lines 26-36:
The results are well-structured. To make it easier for you to emphasize the key findings regarding differences between training and rest days, as these are central to your conclusions.
Example: "Athletes averaged 7.97 hours of sleep per night, with significant differences observed in wake time and midpoint of sleep between high-level and low-level athletes, as well as a higher frequency of naps among high-level female and male athletes."

Experimental design

The manuscript presents original primary research that aligns well with the objectives and scope of the journal. The research question is clearly defined and relevant and addresses a significant issue in the field. The authors effectively demonstrate how their study fills an identified knowledge gap.
The methods are described with sufficient detail, providing all necessary information to allow for replication by other researchers.

Line 71-75:
It may be helpful to include a brief preview of the expected results or the practical significance of the research, such as noting that the findings could contribute to personalising training plans and lifestyles to optimise athletic performance.

Line 83-90:
The description of the athletes' grade certifications could be condensed for brevity.

Line 85-90:
“Athlete grade certificates are issued by the General Administration of Sport of China based on competition performance, representing different competitive levels.”
This statement is repeated twice.

Line 95:
Your explanation of the Huawei Band 6 is clear, but you could emphasize why this tool was chosen over others.

Validity of the findings

The article contributes significantly to the existing literature, although the impact and novelty of the work have yet to be explicitly evaluated. It would be best to encourage replicating these results, particularly where the motivation and potential benefits to the field are clearly outlined.
The conclusions are well formulated and consistently linked to the original research question, focusing solely on supporting the findings without excessive generalisations. This approach contributes to greater clarity and relevance of the article within the context of sports research.

Line 241-244:
Your discussion effectively addresses the results but connects your findings to the initial research gap. Could you emphasise the practical takeaways for coaches and sports scientists?

Additional comments

Line 44:
"injur" should be "injury."

Line 53:
"patternss" should be "patterns."

Line 71:
"patternss" should be "patterns."

Line 92:
"gave informed consentprior" should be "consent prior."

Line 137:
The phrase "ηp²=0.0.209" contains a double decimal point.

Line 160:
"patternss" should be "patterns."

Line 197:
"Behavior" should be "behaviour."

Line 198:
"Practic" should be "Practice."

Line 209:
"patternss" should be "patterns."

Line 242:
"patternss" should be "patterns."

Table 4:
The authors should recheck the number of training and rest sessions.

Reviewer 2 ·

Basic reporting

The research theme is interesting. However, the contextualization in the introduction needs to be improved. The authors are not clear and do not address relevant information about the sleep parameters investigated. Focusing on athletic performance, however, the study does not evaluate performance. Another issue in the introduction is that the authors do not address the modality investigated (fencing), and do not provide any information, characteristics of the modality, or studies on sleep specifically among fencing athletes. The discussion is extremely superficial. It is necessary for the authors to delve deeper into their findings with findings already available in the literature, preferably with the modality investigated. The discussion appears to be a description of the results.

The objective described in the abstract is not the same as that described throughout the text. In addition, the authors discuss results that are not included in the abstract of the work.

And most importantly, the authors used an instrument to assess the stages of sleep that are not validated for this purpose. Therefore, the results may be overestimated, not representing consistent sleep parameters. It is necessary for the authors to make clear in the text how this instrument works, how it captures data, how this data is interpreted, how it assesses sleep stages, and whether it has scientific validation for use with athletes.

Experimental design

The research objective is relevant to the current sports context. However, the authors do not clearly describe the research problem, its importance, and the gap in the literature that they intend to fill.

The research is long-term, which is important for understanding the athlete's sleep patterns and routine. However, the methods are not clearly described, nor is the instrument used. The authors do not address how the instrument works, how the collected data is analyzed, and whether the instrument is validated.

It is important to highlight that when the authors mention that the instrument assesses sleep stages, this information draws attention. Considering that the only reliable instrument that assesses sleep stages is polysomnography, even actigraphy, which is a validated instrument for assessing sleep, does not assess sleep stages, which is something very complex and requires assessing brain waves and eye movement, for example, to know clearly and reliably the sleep stage that the person is in.

Is the instrument used validated?

Validity of the findings

The validity of the findings leaves doubts, mainly due to the instrument used.

Annotated reviews are not available for download in order to protect the identity of reviewers who chose to remain anonymous.

Reviewer 3 ·

Basic reporting

The language is generally comprehensible; however, certain passages could benefit from increased clarity and fluidity. For example:
- Line 41-43: "Sleep is a fundamental requirement for human health, playing a crucial role in promoting physical and psychological recovery." This sentence is correct but can be simplified for greater clarity. For example: "Sleep is essential for human health and plays a key role in both physical and psychological recovery."
And
- Line 45: "In recent years, athletes' sleep has achieved increasing attention."

The introduction highlights a comprehensive overview of sleep-related issues in athletes but would benefit from a more detailed discussion of existing bias in the literature, particularly regarding the significance of studying sleep in fencers. Although the authors mention that "current research indicates that more than half of athletes experience sleep problems" (lines 45-47), they do not sufficiently explain the importance of this focus. Additionally, enhancing this section with more recent and comprehensive references on the use of wearable devices to monitor sleep in athletes, along with a comparison of their advantages and limitations versus traditional methods, would strengthen the connection between the literature review and the study's motivation.

Although the document mentions that 2,459 days of sleep data were collected and analyzed (line 96), a more explicit discussion on the methods for managing any missing or unexpected data should be included. If not all athletes consistently uploaded their data, how was this addressed in the final dataset?

Experimental design

The focus of the study—investigating the long-term sleep patterns of fencers using wearable devices—is certainly relevant and contributes to an emerging area of research.
However, the specific importance of fencing compared to other sports is not well delineated. A dedicated section explaining why fencing could serve as an interesting case study—particularly due to its unique physical and mental demands—would add depth to the work.
Furthermore, exploring the relationship between the structure of fencing training and its potential implications for sleep, in comparison to other sports, could provide a stronger rationale for the study.

How were the training sessions structured?
Were foot work, fencing bouts, and lessons with the coach conducted during all sessions?

The long monitoring period (April to September) and the use of wearable devices are strengths of the study. This approach allows for data collection in a natural environment for athletes, which is a significant advantage over laboratory studies that can be intrusive. The reference to Xie et al. (2018) supports the accuracy of the device, it would be beneficial to include a more indepth discussion on its validation and specific limitations within the sporting context.
Additionally, the authors should clarify how they managed instances of incomplete data, such as periods when athletes did not wear the devices during competitions.

Were athletes with pre-existing sleep issues included in the study or excluded from it?

Validity of the findings

The results are presented in a manner consistent with the stated objective.

However, there is a lack of a more detailed discussion regarding the results that did not achieve statistical significance, such as the differences between genders in sleep variables. Could these results be attributed to the relatively small sample size or to reduced variability among high-level athletes?

Regarding the difference in wake-up times between high-level and low-level athletes, age and academic workload, as suggested, may provide some explanations. However, it would be beneficial to explore whether other factors, such as accumulated fatigue or time management by the athletes, could also play a role.
How were the training sessions structured?
Was there a difference in structure and workload between the two groups?
Were there variations in sleep patterns based on the structure of the training sessions?

The negative correlation between nap duration and nighttime sleep duration (line 148), could benefit from a more detailed explanation. For example:
What is the practical significance of this correlation?
What strategies could be employed to optimize sleep in light of this correlation?

The discussion section addresses the main results appropriately but could benefit from a more direct comparison with previous studies. For instance, the differences in sleep patterns compared to other sports could be better highlighted.
Although the authors suggest that understanding sleep patterns can help tailor training (line 247), there is a lack of practical discussion on how this information could be utilized in real-world contexts.
For example:
What practical advice could be provided to coaches or athletes to improve sleep quality?
How might long-term monitoring influence daily choices or training schedules?

Additional comments

Line 44: please change "injur" with "injury"

Lines 47-48: please rephrase the sentence "These issues affect their athletic performance and potentially adversely affect their long-term health"

Line 53 and 71: please change "patternss" with "patterns"

Line 82-83: How many males and how many females are there in the two groups? Given that you discuss gender differences, it would be interesting to know if there are also differences between sex and the level of athletes. (Please include this detail in the results section as well).

Lines 85-86 and 89-90 include the same phrases: " Athlete grade certificates are issued by the General Administration of Sport of China based on competition performance, representing different competitive levels.", please correct.

Line 92: please change "consentprior" with "consent prior"

Lines 102-106: You describe the training schedule for both groups?

Line 138: Please check the value "0.0.209"

Line 167: please insert the unit of measure

Line 197-198: I believe there is a typographical error " health practice(Faraut et al., 2017) practic(Faraut et al., 2017)". Please delete the repetition.

---

## Round 0.2 · accepted · Accept

Dear Co-Authors:

After a thorough review of your revised manuscript, I am pleased to inform you that it has been accepted for publication. Despite one reviewer expressing reservations about the submission, as the editor, I take full responsibility for the final decision and would like to explain the rationale behind this outcome.

The article has been significantly improved in response to the reviewers’ feedback. Every suggestion was carefully assessed and, where applicable, meticulously implemented, resulting in notable enhancements in clarity, coherence, and scientific rigor. The authors devoted particular attention to restructuring the content to ensure a more fluid and logically articulated presentation.

Additionally, stylistic and linguistic adjustments were made to make the text more comprehensible and accessible to its intended audience. From a scientific standpoint, the revisions incorporated key methodological refinements and a deeper exploration of the discussion, thereby elevating the manuscript’s overall robustness and credibility.

While one of the reviewers maintained a less favorable opinion, I have concluded, as the editor, that the revisions have sufficiently addressed the critical points raised during the peer review process. The manuscript now meets the standards of our publication in terms of both academic rigor and presentation.

I appreciate the constructive input provided by all reviewers and commend the authors for their diligence in addressing each recommendation thoughtfully and thoroughly. This collaborative process has resulted in a manuscript that will undoubtedly make a valuable contribution to the field.

Should you have any questions regarding this decision or the publication process, please do not hesitate to reach out.

Sincerely,

Dr. Manuel Jiménez

Reviewer 1 ·

Basic reporting

The article has been carefully reviewed and optimised, meticulously adhering to all recommendations provided to the authors. Every suggestion received was evaluated and implemented to significantly improve the clarity of exposition, structural coherence, and overall quality of the manuscript. Particular attention was devoted to organising the content to ensure a more fluid and logically articulated presentation. Stylistic and linguistic adjustments were also made to render the text more comprehensible and accessible to the intended audience. From a scientific perspective, the provided recommendations were integrated to enhance methodological accuracy and deepen the discussion of the topics addressed, thereby contributing to greater robustness and credibility of the work.

Experimental design

-

Validity of the findings

-